# Improving model fairness in image-based computer-aided diagnosis

Mingquan Lin [1] ✉, Tianhao Li[2], Yifan Yang [3], Gregory Holste [4], Ying Ding[2], Sarah H. Van Tassel[5], Kyle Kovacs[5], George Shih[6], Zhangyang Wang[4], Zhiyong Lu[3], Fei Wang [1] & Yifan Peng [1] ✉

Deep learning has become a popular tool for computer-aided diagnosis using medical images, sometimes matching or exceeding the performance of clinicians. However, these models can also reflect and amplify human bias, potentially resulting inaccurate missed diagnoses. Despite this concern, the problem of improving model fairness in medical image classification by deep learning has yet to be fully studied. To address this issue, we propose an algorithm that leverages the marginal pairwise equal opportunity to reduce bias in medical image classification. Our evaluations across four tasks using four independent large-scale cohorts demonstrate that our proposed algorithm not only improves fairness in individual and intersectional subgroups but also maintains overall performance. Specifically, the relative change in pairwise fairness difference between our proposed model and the baseline model was reduced by over 35%, while the relative change in AUC value was typically within 1%. By reducing the bias generated by deep learning models, our proposed approach can potentially alleviate concerns about the fairness and reliability of image-based computer-aided diagnosis.

Deep learning has been widely used in healthcare and increasingly demonstrated expert-level performance across various domains[1–7]. However, the issue of fairness has emerged in multiple medical domains and populations[8]. In deep learning, fairness is defined as the absence of prejudice or favoritism toward an individual or group based on their inherent or acquired characteristics[9]. Unfortunately, deep learning models biased by race[10–13], sex[11–15], and age[11–13] have been observed in medical domains. While significant efforts have been made to identify deep learning biases, reducing such biases has been relatively unexplored. Several methods have been proposed to improve group fairness, but they often result in a reduction in model performance[16–19]. In addition, only a few of these methods have been evaluated on relatively large datasets, which may limit their generalizability to real-world scenarios.

In this study, we aim to explore the unfairness issue in using deep learning for image-based computer-aided diagnosis and reduce the model decision bias in underdiagnosed and overdiagnosed patient[12] on the individual and intersectional groups spanning race, sex, age, and genotype. We conducted a comprehensive and systematic analysis to evaluate the effectiveness of our proposed model in reducing unfairness using four publicly available datasets (Fig. 1) designed to detect: COVID-19 from chest X-rays (CXR); thorax abnormality from CXR; primary open-angle glaucoma (POAG) from optic discs; and late age-related macular degeneration (Late AMD) from color fundus photographs (CFP). Our results suggest that model unfairness is pervasive across all large datasets used in image-based diagnosis. Importantly, our proposed model can potentially mitigate the unfairness for both individual and intersectional groups, without affecting the overall performance of the model as measured by the AUC.

[1]Department of Population Health Sciences, Weill Cornell Medicine, New York, USA. [2]School of Information, The University of Texas at Austin, Austin, TX, USA. [3]National Center for Biotechnology Information, National Library of Medicine, National Institutes of Health (NIH), Bethesda, MD 20894, USA. [4]Department of Electrical and Computer Engineering, The University of Texas at Austin, Austin, TX, USA. [5]Department of Ophthalmology, Weill Cornell Medicine, New York, USA. [6]Department of Radiology, Weill Cornell Medicine, New York, USA. ✉e-mail: mil4012@med.cornell.edu; yip4002@med.cornell.edu

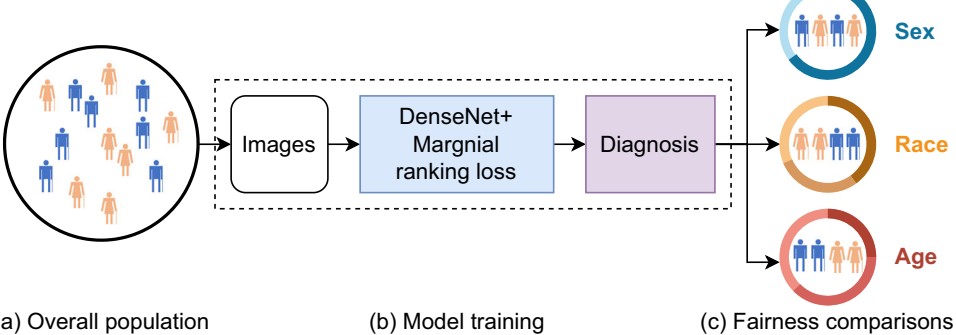

(a) Overall population (b) Model training (c) Fairness comparisons

**Fig. 1 | The model pipeline. a** We used four large-scale publicly available datasets (MIDRC, MIMIC-CXR, OHTS, and AREDS) with a diverse population to detect COVID-19 from CXR, thorax disease abnormality from CXR, primary open-angle glaucoma (POAG) from the optic disc, and late age-related macular degeneration (Late AMD) from color fundus photographs, respectively. **b** We trained a deep learning model with marginal ranking loss using the data specific to each disease. **c** We evaluated pairwise fairness across different subgroups, including sex, race, age, and genotypes, to determine if the model is equally fair for all individuals in each subgroup.

## Results

Our study involved training image-based classifiers to detect diseases and evaluating the model's performance on the overall population and subgroups based on sex, age, race, and genotype. We compared our proposed model, which was trained using the marginal ranking loss, with a baseline deep convolutional neural network trained with binary cross-entropy loss. The detailed pipeline of our proposed model can be found in the "Methods" section.

We assessed the overall performance of the models by measuring the area under the curve (AUC) of the binarized model prediction for the "disease" label (e.g., POAG). To assess the model fairness, we used the pairwise fairness difference (PFD) metric among all the subgroups[20]. Pairwise Fairness for binary classifiers requires that positively-labeled examples be equally likely to be predicted positively, regardless of subgroup[20]. By subtracting the minimum value of Pairwise Fairness from the maximum, one can obtain the PFD. A large PFD indicates significant disparities at the levels of individual or intersectional subgroups and a lack of fairness in the model's predictions.

We assessed the proposed and baseline models trained in four different tasks: MIDRC for COVID-19 detection (CXR, 77,887 images from 27,799 individuals)[21], MIMIC-CXR for thorax abnormality detection[22] (CXR, 212,567 images from 227,827 studies), OHTS for POAG detection[23] (optic disc, 37,399 images from 1636 individuals), and AREDS for Late AMD detection[24] (CFP, 66,060 images from 4566 individuals). Table 1 lists more detailed summary statistics for the datasets, and the "Methods" section provides the complete description per dataset.

### Achieving model fairness in individual subpopulations on age, sex, and race

We find that the proposed model is effective in reducing disparities across all datasets on age, sex, and race (Fig. 2). We were unable to summarize race with Late AMD detection because the Black subgroup in the AREDS dataset has too few members to be studied reliably (<3.7%).

For COVID-19 detection on the MIDRC dataset, our proposed methods obtained a lower PFD and comparable AUC on age and sex, while obtaining lower PFD and AUC on race compared to the baseline model (Fig. 2a). In addition, we observed from Supplementary Tables 1–3 that male individuals, individuals over 75, and Other races individuals have lower AUC than their counterparts. This indicates that individuals in these groups are more likely to be misdiagnosed than other groups.

**Table 1 | The characteristics of four datasets: MIDRC[21], AREDS[24], OHTS[23], and MIMIC-CXR[22]**

| Disease (Dataset) | Subgroup | Attribute | Positive | % | Total |
|---|---|---|---|---|---|
| COVID-19 (MIDRC) | | No. of images | 39,369 | 50.55 | 77,887 |
| | Age | <75 yrs | 34,328 | 52.38 | 65,542 |
| | | > = 75 yrs | 5531 | 44.80 | 12,345 |
| | Sex | Male | 22,395 | 51.04 | 43,880 |
| | | Female | 16,974 | 49.91 | 34,007 |
| | Race | White | 14,355 | 37.33 | 38,457 |
| | | Black | 21,292 | 70.20 | 30,239 |
| | | Other races | 3722 | 40.50 | 9191 |
| Thorax abnormality (MIMIC-CXR) | | No. of images | 150,509 | 69.19 | 217,536 |
| | Age | <60 yrs | 53,564 | 59.53 | 89,975 |
| | | > = 60 yrs | 96,945 | 76.00 | 127,561 |
| | Sex | Male | 83,823 | 71.16 | 117,790 |
| | | Female | 66,686 | 66.86 | 99,746 |
| | Race | Other races | 132,455 | 70.41 | 188,130 |
| | | Black | 18,054 | 61.40 | 29,406 |
| POAG (OHTS) | | No. of images | 2327 | 6.22 | 37,399 |
| | Age | <60 yrs | 420 | 2.58 | 16,254 |
| | | > = 60 yrs | 1907 | 9.04 | 21,085 |
| | Sex | Male | 1303 | 8.05 | 16,185 |
| | | Female | 1024 | 8.71 | 21,154 |
| | Race | Other races | 1554 | 5.46 | 28,460 |
| | | Black | 773 | 8.71 | 8879 |
| Late AMD (AREDS) | | No. of images | 8521 | 12.90 | 66,060 |
| | Age | <65 yrs | 276 | 7.31 | 3775 |
| | | 65–75 yrs | 3013 | 9.06 | 33,255 |
| | | > = 75 yrs | 5232 | 18.02 | 29,030 |
| | Sex | Male | 3768 | 13.16 | 28,623 |
| | | Female | 4753 | 12.70 | 37,437 |
| | Race | Other races | 8496 | 13.31 | 63,808 |
| | | Black | 25 | 1.11 | 2252 |

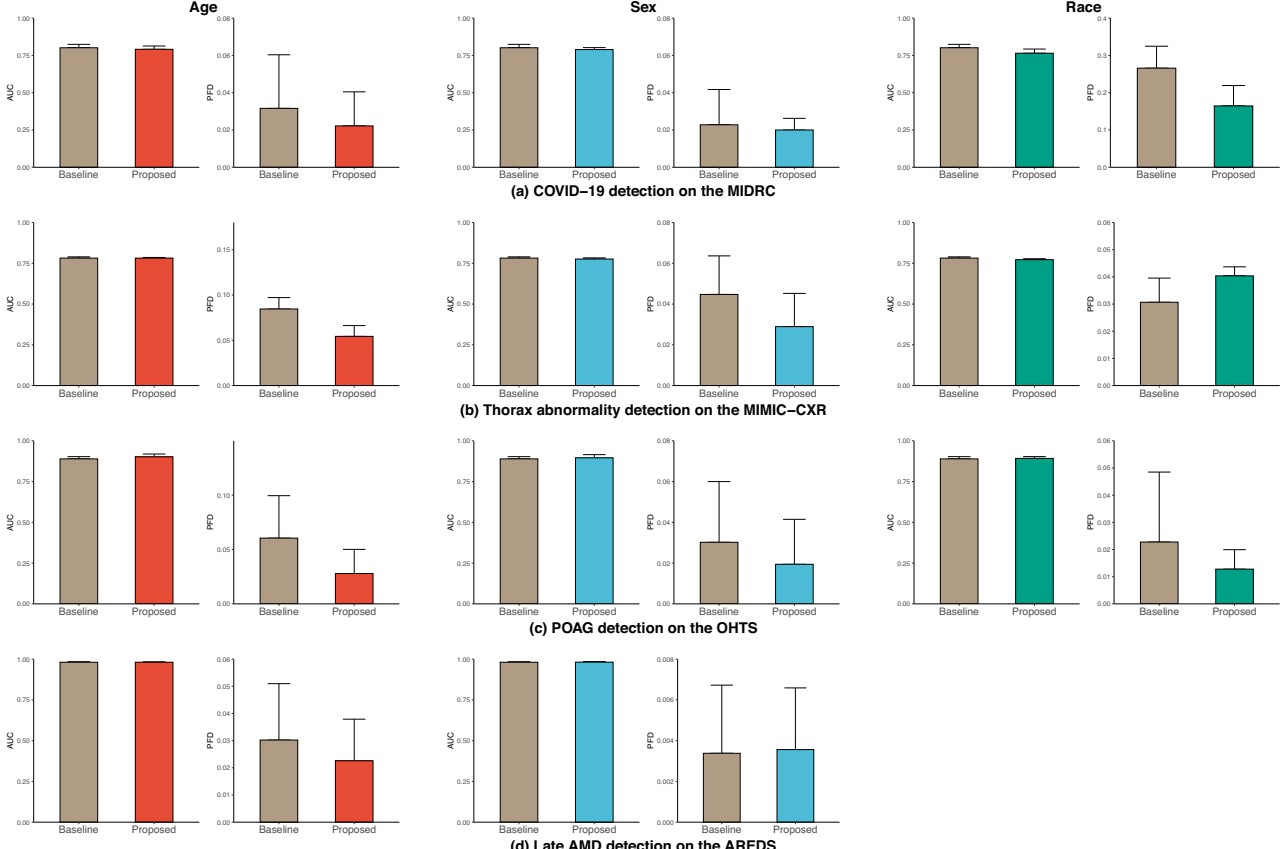

**Fig. 2 | AUC and PFD of DenseNet across subgroups of age (left), sex (middle), and race (right) in the four tasks. a** COVID-19 detection on the MIDRC dataset, **b** thorax abnormality detection on the MIMIC-CXR dataset, **c** POAG detection on the OHTS dataset, and **d** AMD detection on the AREDS dataset. The results are averaged over five trained models using different portions of the data for training (80%) and testing (20%). Standard deviations are also shown. We were unable to summarize race with AMD detection because the Black subgroup in the AREDS dataset has too few members to be studied reliably (<3.7%).

Regarding the thorax abnormality detection on the MIMIC-CXR dataset, the proposed methods achieved lower PFDs and comparable AUCs on age and sex, and slightly higher PFD and comparable AUCs on race compared to the baseline model (Fig. 2b). Supplementary Tables 4–6 revealed that individuals older than 60, males, and Black had lower AUC than their counterparts, suggesting that these groups are more susceptible to thorax abnormality misdiagnosis.

For the POAG detection on the OHTS dataset, our proposed methods yielded lower PFDs and higher AUCs on age, sex, and race than the baseline (Fig. 2c). Supplementary Tables 7–9 showed that individuals younger than 60, female individuals, and Other races individuals exhibit lower AUC than their counterparts, indicating that these groups are more prone to POAG misdiagnosis.

Finally, for Late AMD detection on the AREDS dataset, our proposed methods achieved lower PFD on age and comparable PFD on sex compared to the baseline (Fig. 2d). It is also worth noting that the proposed methods achieved higher AUCs than the baseline. In addition, Supplementary Tables 10–11 showed that individuals younger than 65 had the lowest AUC among all age ranges, while female individuals had comparable AUCs to their male counterparts. These findings suggest that individuals younger than 65 are more susceptible to AMD misdiagnosis.

## Achieving model fairness in individual subpopulations on genotype
We conducted a similar analysis for two genotype groups associated with late AMD and summarized the results in Fig. 3. Supplementary Tables 12–13 provide further details on the reduced disparities achieved by our proposed models for these attributes in the AREDS dataset.

Our proposed methods for the Late AMD detection achieved lower PFD and comparable AUC on CFH and ARMS2 compared to the baseline (Fig. 3). In addition, Supplementary Tables 12 and 13 show that individuals with CFH (TT) or ARMS2 (GG) genotypes had the lowest AUC values, suggesting these groups have a greater likelihood of receiving AMD misdiagnosis.

## Achieving model fairness in intersectional groups
We also investigate intersectional groups, defined as the individuals belonging to two subpopulations, e.g., female Black individuals (Fig. 4). We selected two subpopulations with the largest disparity in pair fairness based on the baseline to form intersectional groups, namely age–race in COVID-19, thorax abnormality, and POAG detection, and age–CFH in AMD detection.

For COVID-19 detection, our proposed methods obtained lower PFD and AUC on the age–sex intersectional group on the MIDRC dataset (Fig. 4a). Supplementary Table 14 shows that the lowest AUC values were observed for younger, Other races individuals, indicating a higher likelihood of misclassification.

Regarding the thorax abnormality detection, the proposed methods achieved a lower PFD and comparable AUC on the age–sex intersectional group on the MIMIC-CXR dataset (Fig. 4b). Supplementary Table 15 shows that female individuals under the age of 60 had lower AUC than its counterpart, suggesting that this group is more prone to thorax abnormality misdiagnosis.

For the POAG detection, our proposed methods achieved a lower PFD and comparable AUC on the age–sex intersectional group than the baseline on the OHTS dataset (Fig. 4c). Supplementary Table 16 further shows that younger female individuals exhibit lower AUC than its

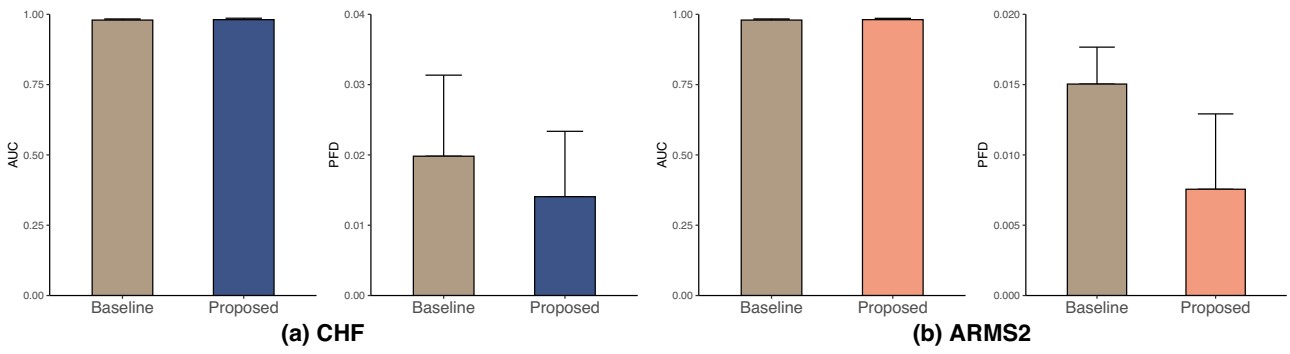

**Fig. 3 | AUC and PDF of DenseNet across subgroups of CHF and ARMS2 associated with late AMD on the AREDS dataset.** The results are averaged over five trained models using different portions of the data to train (80%) and test (20%). Standard deviations are shown. **a** CHF. **b** ARMS2.

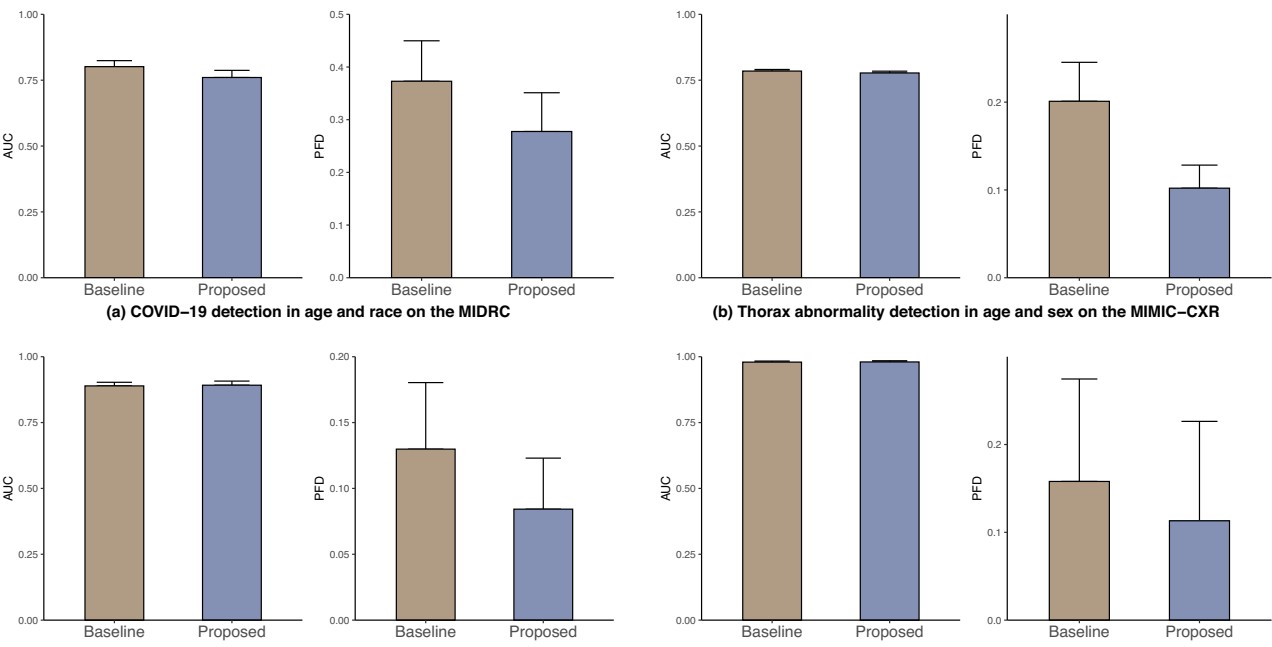

**Fig. 4 | AUC (left) and pair fairness difference (right) of DenseNet across the intersectional groups in the four tasks. a** COVID-19 detection on the MIDRC (age and race), **b** Thorax abnormality detection on the MIMIC-CXR (age and sex), **c** POAG detection on the OHTS (age and sex), and **d** Late AMD detection on the AREDS (age and CFH). The results are averaged over five trained models using different portions of the data to train (80%) and test (20%). Standard deviations are shown.

counterpart, indicating that this group is more prone to POAG misdiagnosis.

Finally, for Late AMD detection, our proposed methods achieved lower PFD and a comparable AUC on age and CFH intersectional group compared to the baseline on the AREDS dataset (Fig. 4d). Moreover, Supplementary Table 17 shows that individuals under the age of 60 with CFH (TT) had the lowest AUC among all age ranges. In contrast, female individuals had comparable AUCs to their male counterparts. These findings suggest that individuals under the age of 60 with CFH (TT) are more likely to have AMD misdiagnosis.

**Evaluating the generalizability and efficacy of the model**
To demonstrate the generalizability and efficacy of our proposed method, we applied ResNet-152, another widely used deep learning model, to two datasets: OHTS and MIDRC. For COVID-19 detection on the MIDRC dataset, our proposed methods obtained a lower PFD and comparable AUC on age, sex, and race compared to the baseline model

(Fig. 5a). In addition, we observed from Supplementary Tables 18–20 that individuals over 75, male individuals, and Other races individuals have lower AUC than their counterparts. This indicates that individuals in these groups are more likely to be misdiagnosed than other groups.

For the POAG detection on the OHTS dataset, our proposed methods yielded lower PFDs and higher AUCs on age, sex, and race than the baseline (Fig. 5b). Supplementary Tables 21–23 showed that individuals younger than 60, female individuals, and Other races individuals exhibit lower AUC than their counterparts, indicating that these groups are more prone to POAG misdiagnosis.

The results generated by ResNet-152 are consistent with those generated by DenseNet-201 based on your proposed method on these two datasets, which suggests the generalizability and efficacy of the proposed method.

We also investigate intersectional groups. For COVID-19 detection, our proposed methods obtained lower PFD and comparable AUC on the age–sex intersectional group on the MIDRC dataset (Fig. 6a).

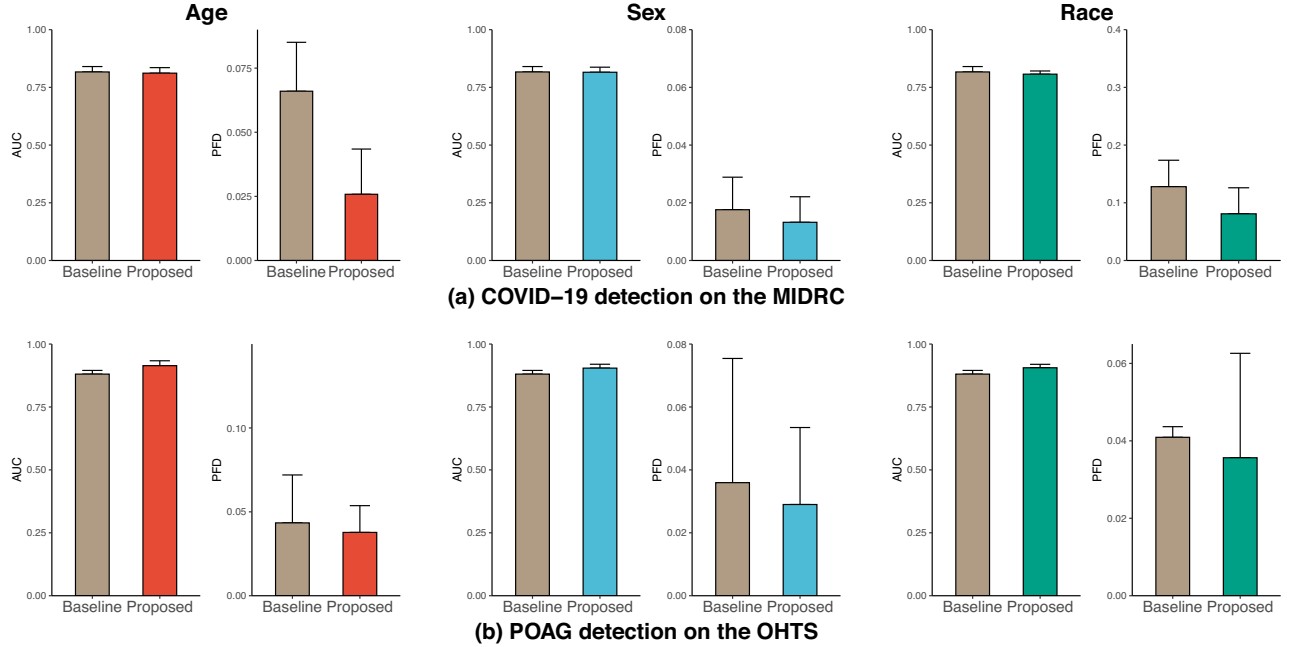

**Fig. 5 | AUC and PFD of ResNet-152 across subgroups of age (left), sex (middle), and race (right) in the four tasks. a** COVID-19 detection on the MIDRC dataset, **b** POAG detection on the OHTS dataset. The results are averaged over five trained models using different portions of the data for training (80%) and testing (20%).

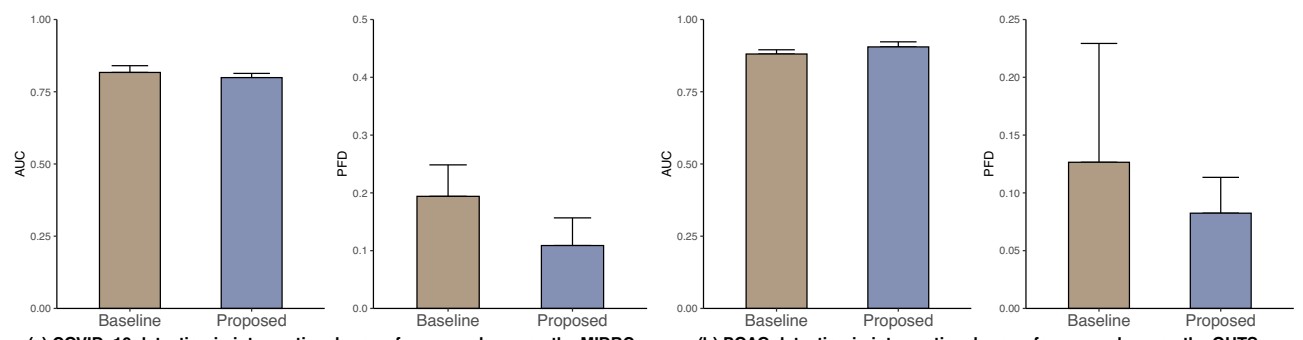

**Fig. 6 | AUC (left) and pair fairness difference (right) of ResNet-152 across the intersectional groups in the two tasks. a** COVID-19 detection on the MIDRC (age and race), **b** POAG detection on the OHTS (age and sex). The results are averaged over five trained models using different portions of the data to train (80%) and test (20%). Standard deviations are shown.

Supplementary Table 24 shows that the lowest AUC values were observed for younger, Other races individuals, indicating a higher likelihood of misclassification.

For the POAG detection, our proposed methods achieved a lower PFD and comparable AUC on the age–sex intersectional group than the baseline on the OHTS dataset (Fig. 6b). Supplementary Table 25 further shows that younger female individuals exhibit lower AUC than its counterpart, indicating that this group is more prone to POAG misdiagnosis.

## Discussion

In this study, we introduce an approach to reduce bias towards groups in deep learning models for image-based computer-aided diagnosis while preserving the overall performance. To evaluate model fairness, we employed the Pairwise Fairness metric, considering it superior to traditional accuracy, sensitivity, and specificity. This choice is rooted in the context of healthcare and clinical decision-making, where risk scores play a pivotal role as decision aids in chronic disease prevention (e.g., POAG and AMD) or health resource triage (e.g., COVID-19). The

proposed model is evaluated on four large-scale datasets for four distinct tasks. In addition, we observed systematic model biases in subpopulations and intersectional groups in all settings. We highlight the following observations for further discussion.

The first observation of our study is that our proposed method effectively improves the fairness of image-based computer-aided diagnosis across different tasks. Compared to the standard binary cross-entropy loss, this method offers two benefits. First, the marginal ranking loss provides a more direct approach to improving Pairwise Fairness by optimizing predictions with incorrect ranking orders, which is particularly effective in cases where samples have lower prediction performance. Secondly, updating the model exclusively using the group with the lowest loss forces the model to learn fairly and consistently improve the lowest Pairwise Fairness across all groups with each batch. These two benefits help achieve Pairwise Fairness for each group and reduce the difference in Pairwise Fairness among all the groups.

To further verify the effectiveness of our proposed method in maintaining AUC and reducing PFD, we conducted a quantitative

**Table 2 | Relative changes in AUC and PDF between baseline and the proposed method in four tasks**

| Disease (Dataset) | Subgroups/Intersec-tional group | Relative change | |
|---|---|---|---|
| | | AUC (%) | PFD (%) |
| COVID-19 (MIDRC) | Age | −2.00 | −40.25 |
| | Sex | −0.96 | −53.79 |
| | Race | −2.44 | −39.73 |
| | Age–Race | −2.54 | −47.69 |
| Thorax abnormality (MIMIC-CXR) | Age | −0.01 | −35.74 |
| | Sex | −0.73 | −35.33 |
| | Race | −1.21 | 31.70 |
| | Age–Sex | −0.92 | −49.24 |
| POAG (OHTS) | Age | 1.42 | −53.82 |
| | Sex | 0.72 | −35.74 |
| | Race | 0.34 | −43.85 |
| | Age–Sex | 2.32 | −35.10 |
| Late AMD (AREDS) | Age | 0.02 | −25.22 |
| | Sex | 0.06 | 5.00 |
| | CFH | 0.15 | −29.06 |
| | ARMS2 | 0.15 | −49.73 |
| | Age–CFH | 0.06 | −28.37 |

analysis using relative change metric[25] (defined in the section "Evaluation metrics"). For PFD, our method outperformed the baseline, with 15 cases showing a decrease, twelve of which were over 35% (Table 2). In addition, most of the relative changes in AUC were within 1%, indicating that our method can achieve a comparable AUC to the baseline. It is worth noting that previous study's enforcement of group fairness constraints always led to a reduction in model performance[16]. To this end, our results demonstrate that the proposed method not only maintains AUC but also improves PFD.

Second, the PFD offers several advantages over conventional fairness metrics, such as equalized odds, demographic parity, and equal opportunity. Previous studies of model fairness often focus on analyzing the disparate impact in binary classification settings, i.e., whether an individual has a particular condition or not. Therefore, these studies often emphasized the disparate impact of binarized metrics such as false negative rates (overdiagnosis rate) and false positive rates (underdiagnosis rate). However, in clinical practice, clinicians also need to make decisions about appropriate resource utilization, and a fairness metric must provide informative rank orderings of individuals. Pairwise Fairness evaluates the problem of bipartite ranking, which ranks positively labeled examples above negative examples between the group and the entire dataset. As a result, it can better capture how the probability of diagnosis is used in clinical practice to inform resource utilization. Furthermore, Pairwise Fairness is scale-invariant, meaning that it only evaluates how well the examples in groups are ranked without using their absolute scores. Therefore, it is classification-threshold-invariant and does not need a threshold to evaluate the model's performance. These advantages make PFD a more suitable fairness metric for clinical decision-making applications.

Third, this study highlights the impact of data imbalance on the bias of deep-learning models. We observed that prevalent patients are overrepresented in some subgroups, leading to biased models. For instance, in the OHTS dataset, the rate of POAG is approximately three times higher in individuals aged ≥ 60 than in those aged <60 (9.04% vs. 2.58%), which can bias the model trained on this dataset. Similar phenomena were observed in the subpopulations of age in the thorax disease detection on the MIMIC-CXR dataset and subpopulations of race in the COVID-19 detection on the MIDRC dataset. Even when

subgroups have similar prevalence, the sample size can still introduce bias. For example, the number of White, Black, and other individuals in the MIDRC dataset are 38,457, 30,239, and 9191, respectively. Although the COVID-19 prevalence is almost the same for the Other races group and White individuals, the model obtained the lowest AUC value for the former group, which had the smallest sample size among the racial subgroups.

Fourth, while data resampling is widely used in pre-processing to mitigate unfairness on subgroups, it may only sometimes be effective. For example, in the POAG detection task on the OTHS dataset, the AUC for female individuals was the lowest among indicated subgroups, despite the comparable number of female and male individuals with POAG. We have also conducted an experiment by oversampling examples from the minority class. The results demonstrate that using an oversampling method can improve model fairness in certain instances, while the proposed method outperforms the oversampling approach (Supplementary Tables 1–17). In addition, obtaining a large number of medical images for biomedical research is often challenging. Generative AI may provide a solution by generating "synthetic" control images using real patients' images and their underlying phenotypes. These "synthetic" images can then be used to develop models, potentially leading to faster and less expensive development of deep learning models for image-based computer-aided diagnostics in new or rare diseases. However, it is crucial to exercise caution in generating images and examining models to avoid further disparate impact.

In addition, we found that model bias is amplified by intersectional attributes compared to individual attributes. The PFD generated by the baseline for the four datasets was over 0.1, which is higher than that for single identities. We also discovered that all intersectional groups were related to the age attribute. This could be due to two reasons. Firstly, the disease could be age-related, as in the case of POAG, where older individuals are more susceptible, and those above 60 years of age tend to have higher AUC values. Secondly, disparities could arise due to data imbalance, where certain groups have smaller sample sizes than others. For instance, in the MIDRC dataset, individuals over 75 had a smaller sample size and lower AUCs than their counterparts.

Finally, the ability to maintain fairness in the presence of uncertainty regarding which subgroup may exhibit bias is a key characteristic of a fairness method. The proposed method performs well in this regard, as it not only generates fair results for subgroups but also further reduces the PFDs even when they are already small. That is demonstrated by comparing the performance of the baseline and the proposed model on the OHTS and AREDS datasets. The PFDs for the race subpopulation in the OHTS dataset and sex subpopulation in the AREDS dataset are 0.0096 ± 0.0257 vs. 0.0074 ± 0.0072 and 0.0034 ± 0.0034 vs. 0.0036 ± 0.0030, respectively. Therefore, the proposed method is both generalizable and applicable in clinical practice.

One limitation of this study is that it only focused on assessing the fairness of binarized models, without examining the calibration of predicted probabilities, which may result in overconfidence or underconfidence in certain cases. Future investigations should explore the relationship between calibration and bias in disease prediction and aim to develop an effective method for reducing calibration bias. In addition, the proposed method can be extended to continuous attributes besides discrete groups and multi-class settings.

In summary, our study proposed a method that effectively reduces unfairness in subgroups in deep learning-based medical image classification, while maintaining overall model performance. The findings revealed evidence of model unfairness in individual and intersectional subgroups across four different disease diagnoses on four datasets, indicating the possibility of such disparities being widespread in other biomedical research. This highlights the importance of addressing these disparities to ensure equitable treatment for

all individuals. The proposed method is shown to be effective in reducing bias while maintaining overall performance, making it suitable for clinical practice, and alleviating concerns about disparities generated by these models.

## Methods

### Data acquisition

In this study, we include four independent datasets (Table 1). These datasets are large and population-based studies, and the research adhered to the principles outlined in the Declaration of Helsinki. In addition, all participants provided informed consent upon entry into the original studies. We use Gen 3 version 2022.10 to download MIDRC and do not use any specific software to download the other three datasets.

The study protocol was approved by the institutional review board at each clinical center and Weill Cornell Medicine. Due to the publicly available nature of both datasets used in this study, the requirement for obtaining written informed consent from all subjects (patients) was waived by the IRB.

**MIDRC**[21]. This dataset is a chest X-ray imaging repository that was specifically created for COVID-19 diagnosis. The repository is part of the Medical Imaging and Data Resource Center (MIDRC)[21], which is a collaborative initiative involving multiple institutions and is funded by the National Institute of Biomedical Imaging and Bioengineering (NIBIB) under contracts 75N92020C00008 and 75N92020C00021 and hosted at the University of Chicago. MIDRC is co-led by the American College of Radiology® (ACR®), the Radiological Society of North America (RSNA), and the American Association of Physicists in Medicine (AAPM). MIDRC accepts images in DICOM format and clinical data in various formats, including COVID-19-related CT scans, X-rays, MRI, and Ultrasound, along with similar control cases. However, for this study, we focused solely on the X-rays. MIDRC collects self-reported race, sex, and age data from its participants. Supplementary Figure 1 provides an overview of the data selection process. In short, the dataset used in this study contains 62,219 imaging studies with demographic information collected up to September 2022. We specifically collected computed radiography (CR) and digital radiography (DX) with age, sex, and race information. Table 1 illustrates the demographic distribution of the imaging studies.

**AREDS**[24]. The Age-Related Eye Disease Studies (AREDS) cohort was a 12-year multi-center prospective study sponsored by the National Eye Institute (National Institutes of Health) that investigated the clinical course, prognosis, and risk factors of age-related macular degeneration (AMD). Between 1992 and 1998, 4757 participants aged 55 to 80 years were recruited from 11 retinal specialty clinics in the United States. The study's inclusion criteria were wide-ranging, from no AMD in either eye to late AMD in one eye. The AREDS dataset is publicly accessible to researchers by request at dbGAP. Comprehensive eye examinations were performed at baseline and annually by certified study personnel using a standardized protocol. Certified technicians captured CFP (field 2, i.e., 30° imaging field centered at the fovea) using a standardized imaging protocol. ADM is classified into early, intermediate, and late stages[26]. Late AMD, the stage characterized by significant vision loss, can manifest in two forms: geographic atrophy (GA) and neovascular AMD (NV). In this study, the focus was on late AMD detection. The ground truth labels were grades previously assigned to each CFP by human expert graders at the University of Wisconsin Fundus Photograph Reading Center. The reading center workflow has been described previously[27]. The dataset includes 66,060 images from 4566 patients, with additional information on self-reported sex and age, as well as two genotypes associated with late AMD, complement factor H (CFH rs1061170) and age-related maculopathy susceptibility 2 (ARMS2 rs10490924). There are 46,244 images

from 2765 patients with CFH and ARMS2. This study did not include race because the Black subgroup in the AREDS dataset has too few members to be studied reliably (<3.7%). More details on the subpopulations are provided in Table 1 and Supplementary Table 26.

**OHTS**[23]. Ocular Hypertension Treatment Study (OHTS) is a large longitudinal clinical trial with 1636 participants and 37,399 images collected from 22 centers in the United States investigating conversion to primary open-angle glaucoma (POAG) in eyes with elevated intraocular pressure. Participants were selected according to eligibility and exclusion criteria[23]. The eligibility criteria include intraocular pressure (between 24 mm Hg and 32 mm Hg in one eye and between 21 mm Hg and 32 mm Hg in the fellow eye) and age (between 40 and 80 years old). The visual field tests were interpreted by the Visual Field Reading Center, and the optic discs at clinical examination and stereoscopic photographs were interpreted by the Optic Disc Reading Center. Exclusion criteria included previous intraocular surgery, visual acuity worse than 20/40 in either eye, and diseases that may cause optic disc deterioration and visual field loss (such as diabetic retinopathy). The gold standard POAG labels were graded by two masked certified readers at the Optic Disc Reading Center, with disagreements resolved by a senior reader. The POAG diagnosis was validated in a quality control sample of 86 eyes (50 normal eyes and 36 with progression), with test-retest agreement at κ = 0.70 (95% confidence interval [CI], 0.55–0.85. More detailed information on the reading center workflow can be found in Gorden et al.[28].

**MIMIC-CXR**[22, 29]. MIMIC-CXR is a large public dataset of 377,110 chest X-rays associated with 227,827 patients presenting to the Beth Israel Deaconess Medical Center Emergency Department between 2011 and 2016. Labels were derived from an open-source labeler tool, CheXpert[30]. In this study, we only used 212,567 CXR Posterior-Anterior and Anterior-Posterior images from 227,827 studies. The race and sex data were self-reported in the MIMIC-CXR dataset, and age was reported at the time of a patient's first admission.

### Pairwise fairness

To assess the fairness of the model, we used the marginal pairwise equal opportunity criterion (Pairwise Fairness)[20]. We deem the pairwise fairness is better than traditional accuracy, sensitivity, and specificity, because, in healthcare and other clinical decision-making settings, risk scores are used as decision aids for the prevention of chronic disease (e.g., POAG and AMD)) or triage of health resources (e.g., COVID-19). Specifically, the criterion measures the "Area under the ROC Curve (AUC)" for a subgroup by calculating the probability that the model ranks a randomly selected positive sample from the subgroup higher than a randomly selected negative example in the entire data:

$$\text{Pairwise Fairness} := P(f(x) > f(x') | y > y', (x,y) \in G_i^+, (x',y') \in G^-) \quad (1)$$

$G$ is the dataset used, $G_i$ is the subgroup in the dataset, $f(x)$ is the output of the AI model with input $x$, and $y$ is the ground truth label of $x$.

### Model development

The pipeline of the proposed model is depicted in Fig. 1. The input images are passed into a convolutional neural network, which generates prediction results. The proposed method is not exclusive to specific deep learning models, and DenseNet-201 showed good performance in classification on OHTS and MIMIC-CXR in our previous study[5, 31]. Therefore, we used the DenseNet-201[32] pretrained on ImageNet[33] for ARDES, OHTS, and MIMIC-CXR datasets, and DenseNet-121[32] pretrained on CheXpert[3] for the MIDRC dataset in this study. To demonstrate the generalizability and efficacy of our proposed method, we applied ResNet-152[34], another widely used deep learning model, to

two datasets: OHTS and MIDRC. We replaced the last layer with a new randomly initialized, fully connected layer with 2 output neurons (abnormal and normal). To achieve fairness among the subgroups, instead of using binary cross-entropy loss, we propose a method to optimize the marginal ranking loss of the group with the lowest Pairwise Fairness. In the training procedure, for each batch in an epoch, we calculated Pairwise Fairness for each subgroup and selected the subgroup with the lowest Pairwise Fairness to calculate the margin ranking loss:

$$\mathscr{L} = \frac{1}{n} \sum \max(0, -x_p + x_n + margin) \quad (2)$$

$x_p$ is the prediction of a random sample with a positive ground truth label for that subgroup. $x_n$ is the prediction of a random sample with a negative ground truth label for the whole training data. $n$ is the number of all the possible pairs of $x_p$ and $x_n$. $margin$ is the threshold that determines when the ranking order of $x_p$ and $x_n$ is considered incorrect or very similar. Compared to previous approaches, the proposed loss function can directly optimize predictions that have incorrect ranking orders, especially for samples with the lowest Pairwise Fairness. In addition, it can update the model by focusing on the group that experiences the least Pairwise Fairness loss, thereby improving the model's performance on that specific group. This approach encourages fair learning and promotes consistent improvement across all groups.

### Evaluation metrics

We reported the average AUC and the difference between the maximum and minimum values of the Pairwise Fairness (PFD). We also used relative change to quantitatively analyze changes in the AUC and PFD obtained by baseline and the proposed:

$$\text{Relative change} := \frac{x_{\text{proposed}} - x_{\text{baseline}}}{x_{\text{baseline}}} \quad (3)$$

where $x_{\text{baseline}}$ and $x_{\text{proposed}}$ are the results obtained by the baseline and proposed model.

### Experimental settings

For the MIDRC dataset, a method similar to Johnson et al.[22] was used to process the original CXRs. First, each Posterior-Anterior (PA) or Anterior-Posterior (AP) CXR was converted from DICOM to JPG format. The pixel values were then normalized to [0, 255] and then inverted, if necessary, to make the air in the CXR white. Histogram equalization was then applied to enhance the image's contrast. Finally, the processed image was saved as a JPG with a quality factor of 95. For other datasets, the images were already in JPG format.

All the images were then resized to $224 \times 224 \times 3$. The network was optimized using the Adam[35] optimization algorithm with a learning rate of $10^{-4}$. The batch size is set to 96. To augment the data, random rotations and flips were applied to the images, with the rotations between $0°$ and $10°$ and horizontal or vertical flips. The experiments were performed on an Intel Core i9-9960 X cores processor and NVIDIA Quadro RTX 6000 GPU. The proposed model was trained for 20 epochs and the model with the highest AUC in the development set was saved. The models were implemented using PyTorch[36].

For the MIDRC, ARDES, and OHTS datasets, the entire dataset was randomly split at the patient level. One group (20% of the total subjects) was used as the hold-out test set and the remaining as the training set. For the MIMIC-CXR dataset, the official release training, validation, and testing datasets were used. All experiments were repeated five times to obtain the distribution of the evaluation metrics.

### Reporting summary

Further information on research design is available in the Nature Portfolio Reporting Summary linked to this article.

### Data availability

The MIDRC dataset used in this study are available in the Medical Imaging and Data Resource Center database [https://data.midrc.org/explorer]. The AREDS dataset used in this study are available in the NCBI dbGAP database under accession code phs000001.v3.p1. The OHTS data are available under restricted access for patient protection. Access can be obtained by requesting (https://ohts.wustl.edu/). The MIMIC-CXR dataset used in this study are available in the PhysioNet database [https://www.physionet.org/content/mimic-cxr-jpg/]. All data supporting the findings described in this manuscript are available in the article and in the Supplementary Information and from the corresponding author upon request. All source datasets are public datasets that can be accessed based on the links in this paper. Source data are provided with this paper.

### Code availability

Codes are available at https://doi.org/10.5281/zenodo.8226443.

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

## Acknowledgements

This work was supported by the National Library of Medicine under Award No. 4R00LM013001 (Y.P.), the National Eye Institute under Award No. R21EY035296 (Y.P., F.W., S.H.VT), National Science Foundation under Award No. IIS-2212176 (Z.W.) and CAREER Award No. 2145640 (Y.P.), Intramural Research Program of the National Institutes of Health, National Library of Medicine (Z.L.), Amazon Research Award (Y.P., G.S.), and the UT Good System project "Being Watched: Embedding Ethics in Public Cameras" (Z.W.).

## Author contributions

M.L. implemented the methods, conducted the experiments, and wrote the paper. T.L. implemented the methods, conducted the experiments. Y.Y., G.H., Y.D., S.H.VT., K.K., G.S., Z.W., Z.L., and F.W. edited the paper. Y.P. advised on all aspects of the work involved in this project and assisted in the paper writing. All authors read and approved the final manuscript.

## Competing interests

The authors declare no competing interests.
