## [Peer Review File · Nature Communications]

Improving model fairness in image-based computer-aided diagnosisREVIEWER COMMENTS

Reviewer #1 (Remarks to the Author):

The idea of improving model fairness in image-based computer-aided diagnosis is important. Overall the paper is nicely written and easy to follow. However, more experiments and justifications are needed to better establish how and why the proposed approach standouts.

The authors will find the below comments useful in enriching the manuscript:

1) It is not clear what specific bias the authors are addressing in this study: Implicit Bias (e.g., confirmation bias) or bias due to improper sampling or imbalance in the dataset.

From the sentence -'this study highlights the impact of data imbalance on the bias of deep-learning models' - at the Discussion section it is arguably understandable that they are focusing on bias due to imbalance in the data, however, it needs to be better articulated in the Introduction section.

2) They are proposing a loss function to handle imbalanced data. Does it outperform the very naïve strategy (i.e., oversampling the training data)? This need to be experientially evaluated.

3) A list of methods (including a list of loss functions) are already available aiming to address data imbalance problem. A detailed justification/experimental evaluation is needed to establish the reason about why/how the proposed method standouts.

4) Only AUC has been used for evaluation. What about accuracy, sensitivity and specificity?

5) By AI, they are only focusing on deep learning (e.g., CNN) models?? It also needs to be better articulated.

6) What is the reason behind sticking to DenseNet-201 only?

Reviewer #2 (Remarks to the Author):

The paper studies fairness in image-based diagnosis. More specifically the paper proposes a method to improve fairness according to pairwise fairness notion and demonstrates its success on different datasets and imaging domains. The paper studies a very important and critical issue. The paper performs reasonable amounts of studies and experiments. Although the paper is written in a language that is easy to follow, the paper is not well organized. Some of the methodological discussions can also be made more clear in the paper along with discussion of more related work. I highlight some of the examples on how to improve the paper according to these two concerns below:

I would suggest the methods section to be discussed before the Results section. It is really strange to talk about the results before talking about methodological details. Also, although Figure 1 is discussed in the beginning of the paper that provides a high-level overview of the method, some components of it are not clear until one goes and reads section 4, so I would suggest the authors to consider this and do some re-organization.

Although it is expected to be a known fact, but it would be good if a citation or two is added here to back up the following claim "Deep learning has become a popular tool for computer-aided diagnosis using medical images, sometimes matching or exceeding the performance of clinicians.". Overall, I think the paper can discuss related work more comprehensively and provide more references in the body of the work both acknowledging past work and as evidence to claims made in the paper.

For the methods' section:

1. Authors should discuss why they chose pairwise fairness as their notion/measure of fairness. What is the justification? Why not other measures/definitions?
2. Authors should discuss more in depth why they chose the loss function that they used during training. Although the paper can benefit from some theoretical discussions, it is also enough to provide one or two sentences on why you think this is a good idea and the reason behind your design choices.
3. It would also be good if the approach can be evaluated using other fairness metrics.

A more general question: Do you think in such sensitive applications the loss on AUC is acceptable? What would be the ideal threshold? Some discussion on this can also be beneficial.

Responses to Referees

NCOMMS-23-16715

Title: “Improving model fairness in image-based computer-aided diagnosis”

Authors: Mingquan Lin, Bojian Hou, Swati Mishra, Tianyuan Yao, Yuankai Huo, Qian Yang, Fei Wang, George Shih, Yifan Peng (corr-auth)

Note: Quotes shown in red in this document as well as in the revised manuscript represent revisions or additions to the previous version of the manuscript. Quotes in black are existing text, reproduced here to provide the context of the discussion.

Response to Referee #1

General comments:

The idea of improving model fairness in image-based computer-aided diagnosis is important. Overall the paper is nicely written and easy to follow. However, more experiments and justifications are needed to better establish how and why the proposed approach standouts.

We thank the reviewers for the effort in reviewing this paper and for these positive comments on our paper. The following is the list of revision actions recommended by the referee.

Comment 1. It is not clear what specific bias the authors are addressing in this study: Implicit Bias (e.g., confirmation bias) or bias due to improper sampling or imbalance in the dataset. From the sentence -'this study highlights the impact of data imbalance on the bias of deep-learning models' - at the Discussion section it is arguably understandable that they are focusing on bias due to imbalance in the data, however, it needs to be better articulated in the Introduction section.

In this study, we are addressing the model decision bias in underdiagnosed and overdiagnosed patients (PMID: 34893776). Our experiments indicate that such biases can stem from various

causes such as data sampling biases, societal biases, and, more significantly, compounded biases. We have made these points clear in the introduction.

On page 1.

“In this study, we aim to explore the unfairness issue in using deep learning for image-based computer-aided diagnosis and reduce the model decision bias in underdiagnosed and overdiagnosed patients [12] on the individual and intersectional groups spanning race, sex, age, and genotype.”

2. They are proposing a loss function to handle imbalanced data. Does it outperform the very naïve strategy (i.e., oversampling the training data)? This needs to be experimentally evaluated.

We have performed an experiment involving oversampling the training data, and the results indicate that the proposed method outperforms the oversampling approach. We have added the results in Supplementary Table 1-17.

“We have also conducted an experiment by oversampling examples from the minority class. The results demonstrate that using an oversampling method can improve model fairness in certain instances, while the proposed method outperforms the oversampling approach (Supplementary Table 1-17).”

3. A list of methods (including a list of loss functions) are already available aiming to address data imbalance problem. A detailed justification/experimental evaluation is needed to establish the reason about why/how the proposed method stands out.

Our proposed method aims to improve model fairness rather than address the data imbalance problem. However, during our analysis, we observed that data imbalance could contribute to bias, and we have highlighted this finding. We also discussed the potential bias that can arise even when attributes have similar sample numbers. Based on the results, the proposed method efficiently improves model fairness. In line with comment #2, we conducted an experiment by oversampling examples from the minority class. The results demonstrate that the proposed method outperforms the oversampling approach. Furthermore, the results suggest that oversampling the minority class in the training data may only successfully alleviate unfairness on subgroups in some scenarios. In the discussion section, we delve into other prospective methods, such as data synthesis, to tackle the issue of data imbalance and enhance model fairness.

On page 10-11.

“In addition, obtaining a large number of medical images for biomedical research is often challenging. Generative AI may provide a solution by generating “synthetic” control images using real patients’ images and their underlying phenotypes. These “synthetic” images can then be used to develop models, potentially leading to faster and less expensive development of deep learning models for image-based computer-aided diagnostics in new or rare diseases. However, it is crucial to exercise caution in generating images and examining models to avoid further disparate impact.”

4. Only AUC has been used for evaluation. What about accuracy, sensitivity and specificity?

In this study, we propose to evaluate the model fairness using rank-based pairwise fairness, rather than accuracy, sensitivity, and specificity, because, in healthcare and other clinical decision-making settings, risk scores are used as decision aids for prevention of chronic disease (e.g., POAG and AMD)) or triage of health resources (e.g., COVID-19), where a variety of interventional resource intensities are available. In the revision, we make this point clearer in Section 4.2.

5. By AI, they are only focusing on deep learning (e.g., CNN) models?? It also needs to be better articulated.

In our research, we primarily focus on deep learning techniques, as they have demonstrated top-tier performance across the four benchmarks utilized in our study. For greater accuracy, we have replaced references to "AI" with "Deep learning."

6. What is the reason behind sticking to DenseNet-201 only?

In our study, we selected DenseNet-201 due to its exceptional performance in classification tasks on two datasets used in our study [32,33]. However, our proposed method is not exclusive to DenseNet-201. To demonstrate the generalizability and efficacy of our proposed method, we applied ResNet-152, another widely used deep learning model, to two datasets: OHTS and MIDRC. The results demonstrate that the proposed method can effectively reduce bias while maintaining overall performance (Figures 5 and 6).

Response to Referee #2

The paper studies fairness in image-based diagnosis. More specifically the paper proposes a method to improve fairness according to pairwise fairness notion and demonstrates its success on different datasets and imaging domains. The paper studies a very important and critical issue. The paper performs reasonable amounts of studies and experiments. Although the paper is written in a language that is easy to follow, the paper is not well organized. Some of the methodological discussions can also be made more clear in the paper along with discussion of more related work. I highlight some of the examples on how to improve the paper according to these two concerns below:

We thank the reviewers for the effort in reviewing this paper and for these positive comments on our paper. The following is the list of revision actions recommended by the referee.

I would suggest the methods section to be discussed before the Results section. It is really strange to talk about the results before talking about methodological details. Also, although Figure 1 is discussed in the beginning of the paper that provides a high-level overview of the method, some

components of it is not clear until one goes and reads section 4, so I would suggest the authors to consider this and do some re-organization.

As per the formatting requirements of Nature Communications (<https://www.nature.com/ncomms/submit/article>), we positioned the Results section ahead of the Methods section. However, to enhance clarity in our manuscript, we have included Figure 1. This illustrates the complete model pipeline and directs readers to the Methods section for more extensive details.

Although it is expected to be a known fact, but it would be good if a citation or two is added here to back up the following claim “Deep learning has become a popular tool for computer-aided diagnosis using medical images, sometimes matching or exceeding the performance of clinicians.”. Overall, I think the paper can discuss related work more comprehensively and provide more references in the body of the work both acknowledging past work and as evidence to claims made in the paper.

Thank reviewer #2 for pointing this out. We cited references [1-7] to back-up the claims “Deep learning has become a popular tool for computer-aided diagnosis using medical images, sometimes matching or exceeding the performance of clinicians.” In addition, we added more citations in the Introduction section to acknowledge past work.

On page 2

“Several methods have been proposed to improve group fairness, but they often result in a reduction in model performance [16-19].

For the methods’ section:

1. Authors should discuss why they chose pairwise fairness as their notion/measure of fairness. What is the justification? Why not other measures/definitions?

Please refer to our responses to reviewer 1’s comment #4.

2. Authors should discuss more in depth why they chose the loss function that they used during training. Although the paper can benefit from some theoretical discussions, it is also enough to provide one or two sentences on why you think this is a good idea and the reason behind your design choices.

We have added the following text to the Method section.

On page 17

“Compared to previous approaches, the proposed loss function can directly optimize predictions that have incorrect ranking orders, especially for samples with the lowest Pairwise Fairness. In addition, it can update the model by focusing on the group that experiences the least Pairwise Fairness loss, thereby improving the model’s performance on

that specific group. This approach encourages fair learning and promotes consistent improvement across all groups.”

3. It would also be good if the approach can be evaluated using other fairness metrics.

We thank the reviewer for this comment. Please refer to our response to reviewer 1’s comment #4.

A more general question: Do you think in such sensitive applications the loss on AUC is acceptable? What would be the ideal threshold? Some discussion on this can also be beneficial.

To the best of our knowledge, there is no universally accepted ideal threshold at present. In our view, enhancing model fairness should not result in significant drops in the relative changes in AUC. In our study, most relative changes in AUC achieved by the proposed method compared to the baseline, fall within 1%. On the other hand, 10 out of 15 decreases in PFD are greater than 35% (Table 2). These results show that our proposed method can improve model fairness without compromising the overall performance.

REVIEWERS' COMMENTS

Reviewer #1 (Remarks to the Author):

I am happy with the revision.

Reviewer #2 (Remarks to the Author):

I thank the authors for providing responses to my questions. Although most of my questions were addressed, I am still unsure about the questions that both me and R1 have pointed out in terms of other fairness metrics. I wish that discussion was reflected as some sort of an experimental result. With that being said since most of my questions were answered, I feel the paper can be ready for publication.

Responses to Referees

NCOMMS-23-16715

Title: “Improving model fairness in image-based computer-aided diagnosis”

Authors: Mingquan Lin, Tianhao Li, Yifan Yang, Gregory Holste, Ying Ding, Sarah H. Van Tassel, Kyle Kovacs, George Shih, Zhangyang Wang, Zhiyong Lu, Fei Wang, Yifan Peng (corr-auth)

Note: Quotes shown in red in this document as well as in the revised manuscript represent revisions or additions to the previous version of the manuscript. Quotes in black are existing text, reproduced here to provide the context of the discussion.

Response to Referee #1

General comments:

I am happy with the revision.

We thank the reviewers for the effort in reviewing this paper.

Response to Referee #2

I thank the authors for providing responses to my questions. Although most of my questions were addressed, I am still unsure about the questions that both me and R1 have pointed out in terms of other fairness metrics. I wish that discussion was reflected as some sort of an experimental result. With that being said since most of my questions were answered, I feel the paper can be ready for publication.

We thank the reviewers for the effort in reviewing this paper and for these positive comments on our paper. The following is the list of revision actions recommended by the referee. We have added more discussion about the evaluation metrics in the discussion section.

“To evaluate model fairness, we employed the Pairwise Fairness metric, considering it superior to traditional accuracy, sensitivity, and specificity. This choice is rooted in the context of healthcare and clinical decision-making, where risk scores play a pivotal role as decision aids in chronic disease prevention (e.g., POAG and AMD) or health resource triage (e.g., COVID-19).”